First genomic insights into members of a candidate bacterial phylum responsible for wastewater bulking

Sekiguchi Yuji 1 y.sekiguchi@aist.go.jp
Ohashi Akiko 1
Parks Donovan H. 2
Yamauchi Toshihiro 3
Tyson Gene W. 2 4
Hugenholtz Philip 2 5 p.hugenholtz@uq.edu.au
1 Biomedical Research Institute, National Institute of Advanced Industrial Science and Technology (AIST) , Tsukuba, Ibaraki , Japan
2 Australian Centre for Ecogenomics, School of Chemistry and Molecular Biosciences, The University of Queensland , St. Lucia, Queensland , Australia
3 Administrative Management Department, Kubota Kasui Corporation , Minato-ku, Tokyo , Japan
4 Advanced Water Management Centre, The University of Queensland , St. Lucia, Queensland , Australia
5 Institute for Molecular Bioscience, The University of Queensland , St. Lucia, Queensland , Australia
Bishop-Lilly Kimberly
Electronic publication date: 2015 Jan 27
Publication date: 2015
Volume: 3
Electronic Location ID: e740
Received 2014 Oct 14; Accepted 2015 Jan 5
Copyright: © 2015 Sekiguchi et al.
Copyright year: 2015
Copyright holder: Sekiguchi et al.
License: This is an open access article distributed under the terms of the Creative Commons Attribution License, which permits unrestricted use, distribution, reproduction and adaptation in any medium and for any purpose provided that it is properly attributed. For attribution, the original author(s), title, publication source (PeerJ) and either DOI or URL of the article must be cited.
License URL: https://creativecommons.org/licenses/by/4.0/

Keywords: KSB3 phylum, Candidate phylum, Wastewater treatment, Anaerobic biotechnology, Filamentous bulking, Metagenomics

Funding: National Institute of Advanced Industrial Science and Technology Australian Research Council Discovery Outstanding Researcher Award DP120103498 ARC Queen Elizabeth II Fellowship DP1093175 Natural Sciences and Engineering Research Council of Canada The study was supported by the Biomedical Research Institute of the National Institute of Advanced Industrial Science and Technology (AIST). Philip Hugenholtz and Donovan H. Parks were supported by an Australian Research Council (ARC) Discovery Outstanding Researcher Award (DORA) grant DP120103498. Gene W. Tyson was supported by an ARC Queen Elizabeth II Fellowship, grant DP1093175. Donovan H. Parks was also supported by the Natural Sciences and Engineering Research Council of Canada. The funders had no role in study design, data collection and analysis, decision to publish, or preparation of the manuscript.

==============================
Filamentous cells belonging to the candidate bacterial phylum KSB3 were previously identified as the causative agent of fatal filament overgrowth (bulking) in a high-rate industrial anaerobic wastewater treatment bioreactor. Here, we obtained near complete genomes from two KSB3 populations in the bioreactor, including the dominant bulking filament, using differential coverage binning of metagenomic data. Fluorescence in situ hybridization with 16S rRNA-targeted probes specific for the two populations confirmed that both are filamentous organisms. Genome-based metabolic reconstruction and microscopic observation of the KSB3 filaments in the presence of sugar gradients indicate that both filament types are Gram-negative, strictly anaerobic fermenters capable of non-flagellar based gliding motility, and have a strikingly large number of sensory and response regulator genes. We propose that the KSB3 filaments are highly sensitive to their surroundings and that cellular processes, including those causing bulking, are controlled by external stimuli. The obtained genomes lay the foundation for a more detailed understanding of environmental cues used by KSB3 filaments, which may lead to more robust treatment options to prevent bulking.

Introduction

Anaerobic digestion is a major type of biological treatment extensively used around the world (Ahring, 2003a) that is not only cost effective for treating organic waste and wastewater, but also can frequently produce energy in the form of methane (biogas) (Angelidaki et al., 2011). Over the last thirty years, a set of high rate anaerobic digestion reactor configurations have been developed, of which the upflow anaerobic sludge blanket (UASB) technology is the most successful and commercialized configuration (Kleerebezem & Macarie, 2003; van Lier, 2008). Despite the success of this technology, serious performance issues have emerged such as the sudden washout of granular sludge biomass due to overgrowth of filamentous bacteria (bulking), which can lead to complete loss of performance.

Bulking of anaerobic digestion systems can be caused by a variety of filamentous microorganisms (Hulshoff Pol et al., 2004; Li et al., 2008; Yamada & Sekiguchi, 2009) and a phylogenetically novel filament was previously reported to be the cause of bulking in an industrial UASB reactor treating sugar manufacturing wastewater (Yamada et al., 2007; Yamada et al., 2011). Small subunit ribosomal RNA (16S rRNA) gene-based analyses of the bulking sludge (Yamada et al., 2007) revealed that the dominant filament type belongs to candidate bacterial phylum KSB3, originally proposed by Tanner et al. (2000) based on an environmental 16S rRNA gene clone sequence obtained from a sulfur-rich marine sediment (Tanner et al., 2000). Fluorescence in situ hybridization (FISH) with KSB3-specific 16S rRNA-directed probes revealed that the KSB3 filaments are localized at the outer layer of healthy granules (Yamada et al., 2007) which become substantially thicker during bulking. The study of filamentous KSB3 bacteria will undoubtedly contribute to our understanding of and ability to prevent bulking in anaerobic wastewater treatment systems, but has been hampered by an inability to obtain a pure culture despite repeated and long term isolation efforts (Yamada et al., 2011). However, culture-independent molecular and imaging methods are beginning to provide clues regarding the ecophysiology of these organisms. This includes their ability to uptake simple carbohydrates, particularly maltose and glucose, under anaerobic conditions and from these observations it was proposed that high carbohydrate loading in the UASB reactor may trigger proliferation of KSB filament populations (Yamada et al., 2011).

Here, we obtained near complete genomes from in situ populations of the dominant bulking KSB3 filament type and a second moderately related low abundance KSB3 filament via differential coverage binning (Albertsen et al., 2013) using metagenomic data previously reported from a full-scale UASB reactor (Soo et al., 2014). Differential coverage binning groups together anonymous metagenomic fragments (contigs) belonging to the same population based on the similarity of their sequencing coverage across multiple related metagenomes (Albertsen et al., 2013). These genomes represent the first genomic information for candidate phylum KSB3 and provide insights into the metabolism of KSB3 filaments and their ability to cause bulking.

Methods

Samples

Methanogenic sludge samples reported in a previous study (Soo et al., 2014) were used in the present study for shotgun sequencing and fluorescence in situ hybridization (FISH). Briefly, two sludge samples (A1 and A2) were taken from the system at different sampling dates (A1, 25th December, 2012; A2, 16th September, 2010), and sample A1 was further separated into two parts (flocculant sludge [F1] and granular sludges [G1]) by gravimetric settlement (Soo et al., 2014) (Table S1). Each sample had been divided into two parts: one used for obtaining DNA via bead-beating and phenol chloroform extraction and the other fixed in 4% paraformaldehyde for FISH (Soo et al., 2014).

Fluorescence in situ hybridization

KSB3-specific FISH probes were designed in ARB v5.5 (Ludwig et al., 2004) using 16S rRNA genes identified in the KSB3 genomes and KSB3 16S rRNA gene sequences available in the current Greengenes database (May 2013 version) (McDonald et al., 2012). In order to maximize the specificity and fluorescence intensity of the probes, helper probes were also designed (Table S5). FISH was performed as described previously (Sekiguchi et al., 1999) using the probes listed in Table S5 and all probes were hybridized overnight. Probes were labeled with either Alexa488 or Cy3 fluorophores, and probes with different fluorophores were used together for dual-staining FISH. Images were captured using an epifluorescence microscope (Axioplan 2; Carl Zeiss) equipped with a cooled charge-coupled device (CCD) camera (DP72; Olympus) and subsequently processed using imaging software (DP2-BSW, version 2.2; Olympus). Super-imposed images were generated using Adobe Photoshop (CS5.1; Adobe).

Figure 1 Phylogenetic structure of the Modulibacteria (KSB3) phylum based on comparative analysis of 16S rRNA gene sequences, and imaging of KSB3 cells.

(A) Maximum-likelihood phylogenetic tree (RAxML) of public data (accession numbers shown) and the 16S rRNA sequence determined in this study for UASB14. Sequences from the bacterial phyla Nitrospirae, Tenericutes, and Chloroflexi were used to root the tree (not shown). Reproducible interior nodes are indicated as a black circle (>90% bootstrap support for neighbor-joining [NJ], maximum parsimony [MP], and maximum-likelihood [ML] inferences), open circle (>80% support); or open rectangle (>70% support). Nodes without symbols were not reproducible between trees. The scale bar represents 5% estimated sequence divergence. Class-level clades are bracketed to the right of the figure in black. The target ranges of KSB3-specific FISH (fluorescence in situ hybridization) probes used in this study are indicated by colored brackets with the colors corresponding to cell color in (B) and (D). (B) 16S rRNA-targeted FISH detection of UASB14 and UASB270 filaments in the UASB sludge. The abundant UASB14 filaments are labeled green and the low abundance UASB270 filaments are labeled red. (C) Total KSB3 filament abundance highlighted by a phylum-level FISH probe relative to (D) all cells present in the same field (phase-contrast image). Bars in (B–D) represent 10 µm.

Metagenome sequencing

Previously sequenced paired-end and mate-pair metagenomes for samples A1, A2, F1, and G1 (Soo et al., 2014) were supplemented with additional data generated in this study using the same DNA. Paired-end Nextera libraries were prepared for each sample according to the manufacturer’s instructions, quantified using the QuantIT kit (Molecular Probes) and sequenced (2 × 250 bp paired end) on an Illumina MiSeq using the Reagent Kit v2 (Illumina) at the National Institute of Advanced Industrial Science and Technology, Japan (AIST; Table S1). The extra metagenome sequencing generated 7.3 Gb, 13.1 Gb, 1.9 Gbp, and 7.9 Gbp for A1, A2, F1, and G1 samples, respectively (yielding a total of 56.2 Gb for all samples combined). For scaffolding, two additional large-insert mate-pair libraries (∼3 kbp and ∼7.5 kbp) were constructed from sample A1 using the Mate Pair Library Preparation Kit v2 (Illumina) and sequenced on Illumina MiSeq system (MiSeq Reagent Kit v2) yielding 2.3 Gb and 3.0 Gb, respectively (for a total of 7.3 Gb from sample A1 when combined with mate-pair data; Soo et al., 2014).

Community profiling

16S rRNA gene amplicon sequencing of all UASB sludge samples using the Illumina MiSeq system was previously reported (Soo et al., 2014). Community composition was also examined by extracting all 16S rRNA reads from the metagenome datasets using the closed reference picking script in QIIME v1.6.0 (Caporaso et al., 2010b) with the Greengenes database (de-replicated dataset at 97%, March 2013 McDonald et al., 2012) as reference (otu_ picking_method, uclust_ref; similarity cutoff value, 0.95). All reads from paired end sequencing were quality filtered using a QIIME script (split_ library_ fastq.py) with the following stringent parameters to ensure only high quality reads were included in the analysis: read trimming with a Phred quality threshold of 17 (-q 17) and discarding reads shorter than 50% of the input read length (-p 0.5), and then the quality filtered single-end reads were used for the closed reference picking to generate OTU tables.

Assembly and binning

Metagenome assembly and population genome binning followed the approach previously described (Albertsen et al., 2013). A graphical illustration of the workflow is shown in Fig. S1. Briefly, paired-end metagenome reads in fastq format were merged with SeqPrep (https://github.com/jstjohn/SeqPrep) using default settings and Illumina sequencing adapters were removed. Unmerged reads were quality trimmed and filtered using Nesoni v0.112 (https://github.com/ Victorian-Bioinformatics-Consortium/nesoni) with removing low quality bases from reads with a Phred quality threshold of 17, removing homopolymers reads, and eliminating trimmed reads shorter than 30 bases (clip –quality 17 –homopolymers yes –length 30). The merged and trimmed reads from the four metagenomes were co-assembled using SPAdes v2.5.0 with the following parameters suited to a complex metagenomic assembly: –only-assembler -k 67 –sc. Reads from respective samples were separately mapped to scaffolds using BWA v0.7.4 (Li & Durbin, 2010) with the BWA-MEM algorithm using default parameters. Population genome binning using differential coverage (Albertsen et al., 2013) was performed using GroopM v0.1 (Imelfort et al., 2014) with the initial core formation based on contigs/scaffolds longer than 1,500 bp. Manual refinement of population genome bins, and subsequent recruitment of contigs/scaffolds longer than 500 bp was performed using the GroopM tools.

Identification of conserved marker genes

All contigs/scaffolds in each genome bin were translated into six reading frames, and hmmsearch in HMMER3 (Eddy, 2011) was used to identify 111 single copy marker genes conserved in most bacteria (Dupont et al., 2012), 83 phylogenetically-informative marker genes (Soo et al., 2014), and the 38 marker genes proposed by PhyloSift (Darling et al., 2014). To determine the completeness and contamination of each genome bin, the distribution and number of the 111 conserved single copy marker set was determined using CheckM (Parks et al., 2014) with default settings.

Refinement of population genome bins

Scaffolding of metagenome contigs using the mate-pair data was performed with SSPACE v2.0 (Boetzer et al., 2011). SSPACE was run with the following two sets of parameters: lower stringency for minor population genomes with relatively low coverage (e.g., UASB270), -k 2 (minimum number of links to compute scaffold) -a 0.7 (maximum link ratio between two best contig pairs) -x 0 (no extention of the contigs using paired reads) -p 1 (making .dot file for visualization) and higher stringency for major population genomes (e.g. UASB14): -k 4 -a 0.7 -x 0 -p 1. The resulting dot files were used for visualizing contig connections using Cytoscape v.2.8.1 (Shannon et al., 2003). In addition, Cytoscape attribute files were generated with coverage, length, and bin number (bin name) information for each contig/scaffold. Based on the coverage information and number of connections between contig/scaffolds, external contig/scaffolds are manually added to each bin. In addition, contig/scaffolds with a small number of connections to other contig/scaffolds in their respective bins were excluded. Refined sets of contig/scaffolds were then scaffolded with SSPACE with the following low stringency parameters: -k 2 -a 0.7 -x 0 -p 1. For further refinement, shotgun mate-pair reads were mapped to the newly generated scaffolds using CLC genomic workbench v6.0 (CLC Bio) using default parameters with the exception of a similarity fraction of 0.98 and exported in SAM format. The assembly was visualized using Circos (Krzywinski et al., 2009) and used for manual inspection of the assembly as previously described (Albertsen et al., 2013). Manual correction of misassembly and mis-scaffolding was performed using the microbial genome finishing module in CLC genomic workbench v6.0 (CLC Bio).

Genome tree

Finished bacterial and archaeal genomes were downloaded from IMG (release 4.1) (Markowitz et al., 2014), from which the 38 universally (Darling et al., 2014) or 83 single-copy proteins broadly conserved in bacteria were identified using HMM searches (Soo et al., 2014). To evaluate the robustness of the protein trees (genome trees), four different outgroup taxon configurations (two data sets for 38 marker genes, two data sets for 83 marker genes) were made (Table S3). Homologous proteins obtained from the KSB3 and reference genomes in each taxon configuration were aligned using hmmalign in HMMER3, and subsequently concatenated. A mask was generated for the concatenated alignment using Gblocks (Talavera & Castresana, 2007) with only conserved positions found in more than half of the sequences considered. All tree topologies were tested for robustness using the maximum likelihood methods from FastTree v2.1.7 (with default parameters, JTT model, CAT approximation) (Price, Dehal & Arkin, 2009) and RAxML v7.7.8 (JTT and Gamma models with rapid 100 times bootstrapping) (Stamatakis, 2006). The PHYLIP SEQBOOT module (Felsenstein, 1989) was used to generate 100 resampled alignments and FastTree was used to analyze the resampled alignments (-n 100). A script (CompareToBootstrap.pl) included in the FastTree package was used to compare the original tree to the resampled trees and generate bootstrap values. Generated trees were imported into ARB (Ludwig et al., 2004), where they were rooted, left-hand ladderized using the “beautify” tool and grouped into phylum-level clades. A representative tree (tree no. 1; Table S4) was exported from ARB and visualized using iTOL (Letunic & Bork, 2011).

16S rRNA gene phylogeny

KSB3 related 16S rRNA genes were manually curated using the Greengenes database (version May 2013; McDonald et al., 2012) in ARB (Ludwig et al., 2004). 16S rRNA genes from binned population genomes were aligned with PyNAST (Caporaso et al., 2010a), imported into ARB, and the alignments were manually corrected using the ARB EDIT tool. Sets of taxa (>1,300 nt) were selected in ARB and their alignments were exported applying Lane mask filtering. One set of taxa included representatives across all recognized bacterial phyla to determine the relative position of KSB3 in the bacterial domain (Fig. S6). A second set of taxa included all KSB3 sequences to determine the relative position of the two UASB filament genomes within the KSB3 phylum (Fig. 1A). Neighbor joining trees were calculated from the masked alignments with LogDet distance estimation using PAUP* 4.0 (Swofford, 2003) with 100 bootstrap resamplings. Maximum likelihood trees were calculated based on the masked alignments using RAxML v7.7.8 (GTR and Gamma models + I) with rapid 100 time bootstrapping. Maximum parsimony trees were calculated using PAUP* 4.0. A heuristic search was used with a random stepwise addition sequence of 10 replicates and nearest-neighbor-interchange swapping. Bootstrap analyses on the maximum parsimony trees were run with 100 times resampling for each best tree. Generated trees were re-imported into ARB for visualization.

Genome analysis

The two assembled KSB3 population genomes were initially annotated with PROKKA v1.7 using default settings (Seemann, 2014), and manually curated by comparison to UniRef90 (Suzek et al., 2007), IMG (Integrated Microbial Genomes, finished genomes, release 4.0) (Markowitz et al., 2014), COGs (Clusters of Orthologous Groups) (Tatusov et al., 2000), PFAM (Punta et al., 2012), and KEGG (Kyoto Encyclopedia of Genes and Genomes) (Aoki-Kinoshita & Kanehisa, 2007) databases. Bi-directional best-blast matches were performed for proteins with matches to UniRef90 and IMG using a bit score threshold of 300, and one-way BLASTP matches with a bit score of 60 (Castelle et al., 2013). For COGs, RPS-BLAST against COG PSSMs from the CDD database (Marchler-Bauer et al., 2013) was performed using an e-value cutoff of 0.01, with the top hit retained for each protein domain. The amino acid sequences were also searched for conserved motifs with PFAM (Punta et al., 2012) using HMMR3 (Eddy, 2011) and PfamScan with default settings (with family noise cutoff). Protein domain structure of some gene products were additionally evaluated using InterProScan search (Quevillon et al., 2005). For manual annotation of the KSB3 genomes, we ranked the resulting annotations as follows: bi-directional best-blast matches with UniRef90 and IMG data; one-way matches with UniRef90, IMG, and COGs; PFAM matches; hypothetical proteins (Castelle et al., 2013). For comparison of gene sets with other genomes, we downloaded the full IMG database (release 4.1) containing all genomes in IMG and their annotations (e.g. PFAMs and COGs). In addition, a list of all finished bacterial genomes and associated metadata (e.g. taxonomic affiliation and genome size) was obtained though IMG. Ribosomal RNA copy number was estimated by determining the ratio of average genome coverage to 16S rRNA gene coverage for each KSB3 genome calculated using BWA read mapping. CRISPR loci were identified using CRT v1.2 (Bland et al., 2007). Presence/absence of some gene sets related to cell envelope structure (Albertsen et al., 2013), complex bacterial lifestyle, and adaptability to fluctuating environmental conditions (‘social IQ’) (Sirota-Madi et al., 2010) were evaluated based on IMG annotation for finished genomes and annotated KSB3 genomes, and the resulting abundance matrix was visualized using R and ggplot2. Orthologous proteins between the two KSB3 genomes were identified using pairwise bi-directional best hit BLASTP searches. Glycoside hydrolases were identified using the CAZy database (Lombard et al., 2014) (dbCAN HMMs v3.0, Yin et al., 2012) with HMMER3 (default settings). Transmembrane proteins were predicted using TMHMM Server v. 2.0 (Moller, Croning & Apweiler, 2001).

Gram-staining and gliding motility

Gram-staining for KSB3 filaments was performed based on the method of Hucker, (Doetsch, 1981). Gliding motility of KSB3 filaments was evaluated using fresh sludge samples examined under an epifluorescence microscope (Axioplan 2; Carl Zeiss) equipped with an automatic thermo-control system (Thermo Plate, MATS-55SFG-FT; Tokai Hit). Fresh sludge samples were placed on glass slides, and a cover glass carefully positioned over the sample to minimize exposure to air. To maintain anaerobic conditions on the slides, reducing agents (Na2S and/or L-cysteine) were added to the samples. The temperature of the microscopic stage was maintained at 37 °C, and time-lapse images were recorded with a cooled CCD camera (DP72; Olympus) equipped with the imaging software (DP2-BSW, version 2.2; Olympus). Glucose, maltose, ribose, mannose, galactose, arabinose, raffinose, sucrose, xylose, fructose, lactate, ethanol, propionate, nitrate, nitrite (final concentration of 5–10 mM) and yeast extract (0.1%) were used as candidate stimuli to induce motility. Each potential stimulant was mixed with the cells prior to the observation, or placed at the edge of the glass cover creating a gradient as the stimulant diffused into the sample.

Results

KSB3 populations in the UASB system

Two UASB sludge samples taken two years apart (A1 and A2), and flocular (F1) and granular (G1) fractions derived from sample A1, reported previously (Soo et al., 2014), were used in the present study. The UASB system had a history of periodic bulking caused by KSB3 filaments (Yamada et al., 2007; Yamada et al., 2011). Inspection of 16S rRNA gene amplicon community profiles of these samples (Soo et al., 2014) revealed two KSB3 16S rRNA phylotypes accounting for 4.9 and 3.7% of total sequencing reads from samples A1 and A2, respectively. The dominant phylotype, representing ∼94% of the KSB3 reads, was identical to the previously reported bulking phylotype (clone YM-1, AB218870; Yamada et al., 2007), and the minor phylotype, representing ∼6% of the KSB3 reads, was identical to a low abundance clone detected in the UASB reactor during normal operation (clone SmB78fl, AB266927; Narihiro et al., 2009). The internal transcribed spacer (ITS) region of the bulking phylotype was sequenced to confirm that it was the same strain present in the bulking and normally operating UASB sludge (Fig. S2). Previously reported metagenomes (Soo et al., 2014) and additional shotgun sequencing of samples A1, F1, G1 and A2 were used for recovering high quality draft population genomes of the two KSB3 phylotypes (Table S1). Based on detection of 16S rRNA genes in the shotgun paired-end read datasets, the KSB3 phylotypes comprised up to 10 and 11% of the A1 and A2 metagenomes respectively with the dominant KSB3 phylotype having approximately 10 fold higher abundance than the minor phylotype (Fig. S3), broadly consistent with the amplicon results.

Recovery of KSB3 population genomes

The four metagenomes (59 Gb in total, Table S1) were co-assembled, generating 504,757 contigs/scaffolds (>500 bp) with a combined length of 906 Mb, an N50 of 3 kb and a longest scaffold of 506 kb. Population genomes were recovered from the assembly by exploiting variations in population abundance (coverage) between individual sample metagenomes (differential coverage binning, Albertsen et al., 2013) using the automated binning tool GroopM (Imelfort et al., 2014). The completeness and contamination of the population genomes were estimated by detection of single copy marker gene sets widely conserved in the domain Bacteria (Dupont et al., 2012). Thirty-nine bacterial population genome bins were obtained with >65% completeness (>73/111 markers) and <10% contamination (<11/111, marker genes with >1 copy in a population genome indicate presumptive contamination with another organism). These genomes were refined by tracking mate-pair reads in network graphs to further improve the completeness and reduce contamination of the bins, and to recruit repeat sequences, notably ribosomal RNA operons, which can evade differential coverage binning if present in multiple copies (Albertsen et al., 2013).

We identified 16S rRNA gene sequences in two refined population genome bins (UASB14 and UASB270, Fig. S4) that were identical to the amplicon sequences from the dominant and minor KSB3 phylotypes, respectively (Table 1). Despite careful manual curation of both genomes, their estimated completeness based on 111 conserved single copy marker genes (Dupont et al., 2012) is only ∼93%, and both also have an inferred ∼6% contamination based on these markers. Inspection of the marker genes with no or >1 hit, however, show a high degree of overlap between the genomes, suggesting that these particular genes are either actually absent, duplicated, or laterally transferred based on phylogenetic inference and gene neighborhood (Table S2). This may not be unexpected given the phylogenetic novelty of the lineage. A revised estimate of completeness and contamination based on a prediction that six of the 111 marker genes are absent and five are duplicated is >98% and <2% respectively (Table 1). To estimate the number of rRNA operons in each KSB3 genome, we compared average genomic coverage to 16S rRNA gene coverage, which indicated that UASB14 and UASB270 have three and two rRNA operons respectively (Fig. S5).

Table 1 Features of the Modulibacteria KSB3 population genomes.

Genome bin identifier	UASB14	UASB270	
Candidatus name	Moduliflexus flocculans	Vecturathrix granuli	
Closest environmental 16S clone	YM-1 (AB218870)	SmB78fl (AB266927)	
No. of scaffolds	8	21	
Total length (bp)	7,147,157	8,384,694	
N50	1,183,318	597,372	
GC (%)	50.6	47.2	
Average coverage	278	38	
Genome completenessa	92.8% (103/111)	93.6% (104/111)	
Revised genome completenessa	98.1% (103/105)	99.0% (104/105)	
Genome contaminationa	5.4% (6/111)	6.3% (7/111)	
Revised genome contaminationa	0.0% (0/105)	1.9% (2/105)	
Relative abundance in UASB metagenomes (%)b	9.22	0.40	
No. tRNA genes	54	43	
rRNA genes found in genome	5S, 16S, 23S	5S, 16S, 23S	
Inferred no. of rRNA operonsc	3	2	
No. CDS	5,989	7,048	
No. CRISPR array	4 (125 repeats in total)	5 (550 repeats in total)	
Coding density	84.6 %	84.3 %	
Putative glycoside hydrolasesd			
Cellulase	5 (0.1%)	14 (0.2%)	
Amylase	19 (0.3%)	8 (0.1%)	
Debranching enzyme	3 (0.1%)	2 (0.1%)	
Amino-sugar-degrading enzyme	35 (0.6%)	45 (0.6%)	
Oligosaccharide-degrading enzyme	43 (0.7%)	23 (0.3%)	
Putative protease/peptidased			
Protease	27 (0.5% of total ORFs)	28 (0.4% of total ORFs)	
Peptidase	60 (1.0%)	78 (1.1%)	
Putative environmental signaling system genes			
Transmembrane sensore	135 (2.3%)	114 (1.6%)	
Response regulator containing CheY-like domainf	131 (2.2%)	116 (1.6%)	
Proposed class	Moduliflexia	Vecturatrichia	
Proposed order	Moduliflexales	Vecturatrichales	
Proposed family	Moduliflexaceae	Vecturatrichaceae	
Notes.

a Genome completeness and contamination were estimated based on the presence/absence of a 111 single-copy gene set from Dupont et al. (2012). Revised genome completeness and contamination were calculated based on a revised total of 105 single-copy genes estimated to be present in the KSB3 genomes (Table S2). Numbers in parentheses indicats detected number of genes per total number of each gene set.

b Relative genome abundance for each KSB3 genome was determined based on 16S rRNA gene profiling using shotgun metagenome data (Table S1).

c Number of rRNA operons in the KSB3 genomes were inferred based on relative coverage profiles of KSB3 16S rRNA genes and the genome averages (Fig. S2).

d Counts (% of total ORFs).

e Predicted number of transmembrane sensors based on the possession of a sensor domain (Galperin, 2004) and >1 transmembrane segments (Table S8).

f Number of all two-domain response regulators containing CheY-like domains estimated from PSI-BLAST searches of domain-specific profiles against the protein set described in Galperin (2004) (Table S8).

KSB3 phylogeny and morphology

The relative position of the two KSB3 genomes within the phylum was assessed by comparative analysis of their 16S rRNA gene sequences with publicly available full-length sequences. UASB14 and UASB270 represent two of several major lines of descent in the KSB3 phylum (Fig. 1A). According to Greengenes classification (McDonald et al., 2012), UASB14 belongs to an unnamed class-level lineage and UASB270 is a member of class MAT-CR-H3-D11 for which we propose the names Moduliflexia and Vecturitrichia, respectively (Table 1; Supplemental Information 1). To confirm the status of KSB3 as a candidate phylum, as inferred by 16S rRNA comparative analyses (Yamada et al., 2007, Fig. S6), we constructed phylogenetic trees based on a larger genomic sampling. Two sets of marker genes broadly conserved in all domains of life (38 markers) (Darling et al., 2014) or in Bacteria (83 markers) (Soo et al., 2014) were obtained from the KSB3 genomes and up to 354 publicly available reference genomes (Markowitz et al., 2014). Each gene family was independently aligned and ambiguous and/or non-informative positions removed, and then the filtered alignments were concatenated for maximum-likelihood inference. Four sets of outgroup configurations were used including representatives of all major genomically sampled bacterial phyla (Table S3). The two KSB3 genomes form a robustly monophyletic group in all analyses, and did not reproducibly affiliate with any other phyla (Fig. 2, Fig. S7, Table S4), consistent with the original proposal that KSB3 is a candidate bacterial phylum (Yamada et al., 2007). The average amino acid identity (AAI) between UASB14 and UASB270 is 60.3% (Fig. S8), and supports their assignment to separate classes as it falls within the range of known class-level AAI values (44–61%, Konstantinidis & Tiedje, 2005).

Figure 2 Maximum-likelihood phylogenetic inference of Modulibacteria (KSB3) population genomes among known bacterial phyla.

The tree was constructed using RAxML based on up to 38 marker genes (using taxon-outgroup configuration Config 3, Table S3) and sequences were collapsed at the phylum level except for classes in the Proteobacteria. Ranks are indicated by prefix; p__ (phylum), c__ (class). KSB3 genomes obtained in this study are highlighted in red. Superphyla (Terrabacteria, Patescibacteria, Fibrobacteres-Chlorobi-Bacteroidetes [FCB], and Planctomycetes-Verrucomicrobia-Chlamydiae [PVC]) are highlighted with color ranges. Taxa comprising cultivated representatives are shown in black; taxa with no cultivated representatives are indicated by outline. Reproducible associations (>80% bootstrap values from 100 resamplings) are indicated by dots on interior nodes. Alignments of homologous proteins from archaeal genomes were used to root the tree (not shown). The scale bar represents 10% estimated sequence divergence.

To confirm the filamentous morphology and relative abundance of the KSB3 phylotypes, we designed 16S rRNA-targeted fluorescence in situ hybridization (FISH) probes specific at the phylum and class level and combined them with previously applied KSB3-specific probes (Yamada et al., 2007) (Table S5). We detected only filamentous KSB3 morphotypes in the UASB sludge and these comprised the majority of observed filaments in sample F1 (Figs. 1C–1D). The relative abundance of the two KSB3 phylotypes inferred from both amplicon and metagenome data was also consistent with FISH analyses; that is, filaments belonging to the class Moduliflexia (presumably mostly UASB14) greatly outnumbered those belonging to the Venturitrichia (presumably mostly UASB270) (Fig. 1B), noting that the two filaments were indistinguishable by light microscopy alone (Figs. 1C–1D).

General features of the KSB3 genomes

Both KSB3 genomes are large by bacterial standards, >7 Mb (Fig. S9) and have median GC content, ∼50% (Table 1). Since UASB14 and UASB270 are not close relatives, large genome size may be a characteristic feature of the KSB3 phylum or at least of the two classes that they represent (Fig. 1A). A total of 5,989 and 7,048 open reading frames (ORFs) were identified in the UASB14 and UASB270 genomes, respectively (Table 1). For both genomes, approximately two thirds of the ORFs had a predicted function and the remaining third were hypotheticals. Reciprocal BLASTP best matches between the predicted gene products of the two KSB3 genomes indicate a shared set of 3,296 orthologs, representing approximately half of the gene inventories in each genome. Included in this common set are conserved genes for translation, nucleotide transport and metabolism, and construction of a diderm (Gram negative) cell envelope including lipopolysaccharide synthesis (Fig. S10). We identified a full complement of rRNA and tRNA genes in UASB14, but not UASB270, which were likely missed in the latter genome (Table 1; Table S6). The UASB14 rRNA genes are estimated to be present as three nearly identical operons (Fig. S5) collapsed into a single large repeat during the assembly process. A number of large clustered regularly interspaced short palindromic repeats (CRISPR) were identified in both genomes (Table 1). CRISPR, together with associated cas genes, constitute a recently described defense mechanism against invading foreign DNAs and have been found in a majority of bacterial genera and most Archaea (Sorek, Kunin & Hugenholtz, 2008). A COG category analysis of the KSB3 genomes indicates that both have high relative proportions of carbohydrate metabolism and transport (G) and signal transduction (T) relative to the bacterial average (Fig. 3). More detailed inferred metabolic properties of the two KSB3 genomes are described below.

Figure 3 Relative representation of COG categories by predicted ORFs in the UASB14 and UASB270 genomes.

Global averages and standard deviation (bars) are shown for 2,279 publicly available finished bacterial and archaeal genomes (Markowitz et al., 2014). Statistically significant differences are indicated by percentile of scores for all the available finished bacterial and archaeal genomes.

Strictly fermentative metabolism

Both KSB3 representatives have an incomplete tri-carboxylic acid (TCA) cycle and lack most electron-transport chain complexes including terminal oxidases, indicating a strictly fermentative metabolism (Fig. 4). They encode, however, both superoxide reductase and thioredoxin reductase, suggesting oxidative stress tolerance. Both genomes have a large complement of transporters and enzymes for importing and degrading complex and simple carbohydrates, which can then be fed into a complete glycolysis (Embden-Meyerhof-Parnas) pathway (Fig. 4; COG category G in Fig. 3). Both filaments also are likely capable of hydrolyzing polymers such as cellulose and starch via a range of glycoside hydrolases (Table S7). They redundantly encode four different enzymes for converting pyruvate to acetyl–coenzyme A (acetyl-CoA), namely pyruvate dehydrogenase, pyruvate-formate lyase, pyruvate ferredoxin oxidoreductase, and pyruvate-flavodoxin oxidoreductase. Both generate adenosine triphosphate (ATP) by converting acetyl-CoA to acetate via two enzymes (acetate kinase and phosphate acetyltransferase) commonly found in general fermentative anaerobes (Mai & Adams, 1996), in addition to glycolysis. They may reoxidize NADH produced during glycolysis by converting pyruvate to D-lactate and acetyl-CoA to ethanol (Fig. 4).

Figure 4 Composite metabolic overview of the Modulibacteria (KSB3) genomes based on identified genes and pathways.

Gray indicates elements common to both genomes, while orange and green show elements specific to UASB14 and UASB270, respectively. Both filament types have the genes necessary to produce acetate, ethanol, lactate, and hydrogen (and possibly propionate) as fermentative end products, likely generating energy through the glycolytic Embden-Meyerhof-Parnas (EMP) pathway and the fermentation of amino acids and sugars. Abbreviations: ETF, electron transfer flavoprotein; Fd-ox and Fd-red, oxidized and reduced ferredoxin, respectively; UQ, ubiquinone.

The KSB3 genomes possess a large complement of enzymes for conversion and transport of amino acid and peptides (Fig. 4; COG category E in Fig. 3), including numerous proteases and peptidases (Table 1). Peptide and amino acid degradation in the KSB3 filaments may produce pyruvate, oxaloacetate, succinyl-CoA, and possibly propionyl CoA (Fig. 4). Notably, a complete set of genes for the methylmalonyl CoA pathway was identified in UASB270, suggesting a role in either amino acid degradation, propionate oxidation and/or propionate formation as a fermentative end product in this organism. Some fermentative anaerobes are known to produce hydrogen to scavenge excess electrons generated during metabolism (Sieber, McInerney & Gunsalus, 2012). In both KSB3 genomes, we identified several hydrogenase genes (Fig. 4). By examining domain structure and gene neighbourhoods (Fig. S11), we predict that some of these genes encode catalytic enzymes. This may permit them to engage in syntrophic interactions with hydrogenotrophs, such as methanogens, in the sludge granules (Sieber, McInerney & Gunsalus, 2012). However, based on FISH experiments highlighting KSB3 and archaeal cells, we did not observe a close proximity between the two groups that would facilitate syntrophy (data not shown). Some of the hydrogenase genes are located next to signal transduction genes raising the possibility that they are involved in signal transduction and chemotaxis (Fig. S11).

Sensory capabilities and motility

One of the most striking features of the KSB3 genomes is the presence of extensive regulatory networks, including two-component signal transduction systems (Table 1; Table S8). Signal transduction genes (COG category T) are among the highest represented categories in both genomes (Fig. 3). Two-component systems respond to a broad range of extracellular and intracellular signals, and play a role in many cellular processes including growth, motility, and the cell cycle (Galperin, 2004; Skerker et al., 2005; Kirby, 2009). UASB14 and UASB270 encode 135 and 114 putative transmembrane sensor proteins likely used for environmental signaling (Galperin, 2004), and 131 and 116 putative response regulators containing CheY-like domains, respectively (Table S8). They each contain over 60 methyl-accepting proteins and numerous Che-like chemotaxis proteins (Table S8). Even when compensating for their relatively large genome sizes, both KSB3 genomes possess high proportions of environmental sensory networks compared to other sequenced bacterial and archaeal genomes (Fig. 5; Figs. S12 and S13; Table S9). The high representation of sensory components in the KSB3 genomes is on par with social Myxococcales such as Sorangium cellulosum and Stigmatella aurantiaca,  both of which exhibit complex, self-organizing behavior in response to environmental stimuli (Huntley et al., 2011).

Figure 5 Number of protein domains inferred to be involved in environmental signaling for the two Modulibacteria (KSB3) genomes and finished bacterial and archaeal genomes.

Number of protein domains inferred to be involved in environmental signaling (Table S8) as a function of genome size for the two Modulibacteria (KSB3) genomes (in red) and 2,279 publicly available finished bacterial and archaeal genomes (in blue). The KSB3 filaments have among the highest proportion of signaling domains, only surpassed by members of the Myxobacteria (open blue circles), which are capable of fruiting body formation by contact-mediated signaling.

Sensory capabilities are an important component of a bacterium’s overall social “intelligence” or social IQ (Ben-Jacob et al., 2004), a metric recently proposed based on the abundance of two-component systems, transcription factors, defense mechanisms and transport systems (Sirota-Madi et al., 2010). We determined that the KSB3 filaments have among the highest social IQ scores of any sequenced bacterial and archaeal species to date, scoring particularly well in the two-component and transport system categories (Fig. S14). This suggests that the filaments are sensitive to their surroundings and capable of adaptable behavior in response to changes in their local environment. Key to this adaptability is motility. No genes for flagella production were identified in either KSB3 genome, so to determine if KSB3 bacteria are indeed motile, we observed filaments enriched from UASB granules by wet mount microscopy under a range of conditions. KSB3-specific FISH of samples taken in parallel confirmed that the majority of filaments in these samples were members of the KSB3 phylum (Fig. 1). Initially no motility was observed, therefore based on the metabolic reconstruction of the KSB3 genomes, we added a range of compounds (mostly simple sugars, see ‘Methods’ section) to the edge of the microscope slides to create a gradient that could be sensed by the filaments to stimulate a motility response. We observed gliding motility at rates of between 20 to 30 µm/min only when a glucose or maltose gradient was applied under conditions mimicking the UASB reactor operation (Movie S1). Both KSB3 genomes encode a number of the genes necessary for type IV pili formation (pilB, pilC, pilG, pilT, pilV, and flp pilus assembly protein) that may enable gliding via extension and retraction (Jarrell & McBride, 2008). However, the full gene complement for pili formation (Mauriello et al., 2010) was not detected and the mechanism for KSB3 gliding motility remains to be determined.

Discussion

Despite the biotechnological significance of industrial-scale anaerobic digestion, our understanding of the microbial ecology that underpins these processes is still rudimentary because most microorganisms cannot be cultured and such systems are essentially managed as “black boxes” (Ahring, 2003b; Rivière et al., 2009). Emerging culture-independent molecular techniques such as differential coverage binning of metagenomic data, which allows even low abundance population genomes to be recovered (Sharon et al., 2013; Albertsen et al., 2013), are providing new opportunities to understand and optimize system performance (Vanwonterghem et al., 2014).

Using this approach, we obtained the first population genomes representing candidate bacterial phylum KSB3 (Tanner et al., 2000; Yamada et al., 2007). One of these genomes, UASB14, belongs to a high abundance filament (∼10% of the community; Table 1; Fig. 1) previously reported to be responsible for bulking in an industrial UASB system treating wastewater from sugar manufacture (Yamada et al., 2007). A second genome from the same habitat, UASB270, represents a low abundance (<0.5%) filament only moderately related to the first, i.e., they represent different classes within the KSB3 phylum (Fig. 1). Metabolic reconstruction indicates that both filaments are primary fermenters of sugar and amino acid-containing compounds in the system (Fig. 4), and both have a high “social IQ” based in part on possession of extensive regulatory networks (Table 1; Tables S8 and S9; Fig. S14). These findings support the hypothesis that KSB3 filaments are important primary fermenters in healthy sludge granules (Yamada et al., 2011) and further suggest that the filaments are sensitive to their surroundings and that their cellular processes, such as growth, may be controlled by external signals. Whether these features can be extrapolated to the whole KSB3 phylum, or simply reflect the specialized habitat from which the genomes were obtained, remains to be determined. Environmental surveys suggest that the phylum has a shallow ecological footprint, having been identified in mostly anoxic saline habitats (Fig. 1A), which may indicate that a fermentative metabolism is universal.

The inferred capacity of the filaments to detect physicochemical gradients in their surroundings suggests that they should be motile. Apart from an incomplete gene complement for Type IV pili, no motility mechanism could be identified. However, microscopic observations indicated that the KSB3 filaments are capable of gliding motility in response to applied sugar gradients (Movie S1). Gliding motility is thought to have evolved independently in multiple bacterial lineages, and the molecular mechanisms of gliding are only partially elucidated for a limited number of bacterial taxa (Jarrell & McBride, 2008; Mignot & Kirby, 2008). This is the first report of gliding motility of organisms in UASB sludge granules, which have long been considered to have an organization driven by growth and attachment rather than motility of cells (Liu et al., 2003; Hulshoff Pol et al., 2004). An enhanced sensory system is also likely the key driver of the bulking phenomenon; that is, changes in the UASB reactor such as increases in glucose or maltose concentration trigger outgrowth of the KSB3 filaments (Yamada et al., 2011). It may also explain why repeated attempts to cultivate KSB3 filaments have failed to date (Yamada et al., 2011), because they require specific and possibly complex environmental cues to stimulate growth in axenic culture.

The inference that the KSB3 filaments sense sugars and the observation of a gliding motility response in the presence of a glucose or maltose gradient is consistent with the previous observation of uptake of these sugars by KSB3 filaments (Yamada et al., 2011). Plant operators began monitoring glucose concentration in the UASB reactor influent using a simple urine test strip. No further bulking has occurred to date since keeping influent glucose concentration uniformly low (<200 mg/L) via adjustment of retention times in the acidification pretreatment. A more detailed understanding of environmental stimuli responsible for growth and bulking will be facilitated by the availability of the KSB3 genome sequences which may lead to genome-directed cultivation (Tyson et al., 2005) and other treatment options for bulking.

We propose the names ‘Candidatus Moduliflexus flocculans’ and ‘Candidatus Vecturithrix granuli’ for the two KSB3 filament types represented by the UASB14 and UASB270 genomes respectively, and the phylum name, Modulibacteria, and intermediate rank names (Table 1; Supplemental Information 1).

Conclusions

In summary, this study adds novel genomic ‘foliage’ to the tree of life by reporting the near complete genomes of two phylogenetically diverse members of candidate bacterial phylum KSB3 obtained from an industrial UASB system. Genome-based metabolic reconstruction and experimental observations provide clues to the roles of the KSB3 bacteria in the treatment system including their ability to ferment sugars and chemotactically respond to glucose and maltose gradients, laying the foundations for a detailed understanding of their ecophysiology and role in wastewater bulking.

Supplemental Information

Supplemental Information 1 Supplementary Notes: Description of new taxa

Click here for additional data file.

Figure S1 Flow diagram of the methods used for metagenomics in this study

Click here for additional data file.

Figure S2 Partial 16S rRNA gene and 16S-23S internal transcribed spacer (ITS) regions obtained from bulking sludges and normal sludge granules

Partial 16S rRNA gene and 16S-23S ITS regions obtained from bulking sludges (samples B1 and B2) and normal sludge granules (samples A1 and A2, see the ‘Method’ section) from the same UASB reactor system. Sample B1 is a DNA sample extracted from the sludge at the bulking event reported previously (Yamada et al., 2007) and sample B2 is from another bulking sludge at a different bulking occasion (Yamada et al., 2011). For the four samples, amplification of 16S-ITS regions from the purified DNA preparations was carried out by PCR with Taq polymerase (AmpliTaq Gold PCR Master Mix, Applied Biosystems) according to the manufacturer’s instructions (∼0.1 ng template DNA, 1 × Taq polymerase buffer, 0.2 units Taq polymerase, 0.2 mM of each dNTP and 0.5 mM of each primer in a 10 µl reaction volume). The PCR primers used in the amplification were a KSB3-16S rRNA gene specific primer KSB3-703f (5′-GAG ATC AGG AAG AAC GTC-3′, the same target site of the probe KSB3-703 shown in Table S5) and a universal 23S rRNA gene-targeted primer 23S-115r (5′-SCG GGT TBC CCC ATT CGG-3′, where S represents G or C, and B indicates C or G or T; slightly modified from Lane 1991). The reaction conditions were as follows: initial denaturation at 95 °C for 9 min, followed by 35 cycles of 95 °C for 0.5 min, 60 °C for 0.5 min and 72 °C for 1.5 min. PCR products showing a single band of amplified DNA (∼1.2 kb) were purified with QIAquick PCR Purification Kit (Qiagen). The DNA fragments were cloned into plasmids (pT7Blue T-Vector, Novagen) using DNA Ligation Kit ver.2 (TaKaRa) and ECOS competent E. coli (Nippon Gene) according to the manufacturer’s instructions. Clonal DNAs were prepared by colony PCR from randomly selected recombinants using primers M13M4 (5′-GTT TTC CCA GTC ACG AC-3′) and M13RV (5′-CAG GAA ACA GCT ATG AC-3′), and the PCR products were purified with MinElute 96 UF PCR Purification Kit (Qiagen). Sequencing was conducted with the purified PCR products as templates with primers M13M4 and T7 (5′-TAA TAC GAC TCA CTA TAG GG-3′) using ABI PRISM BigDye Terminator V3.1 Cycle Sequencing Kit and an automated sequence analyzer (3,500 Genetic Analyzer, Applied Biosystems), according to the manufacturer’s instructions. Sequences obtained were analyzed with CLC genomics workbench v 6.5.1 (Qiagen), and gene annotation was done with PROKKA v1.7 with the default settings (Seemann, 2014). Only two distinct sequence types, ITS-1 (accession number: AB933567) and ITS-2 (AB933568), were identified amongst 22 clones analyzed (distribution shown in the table to the right of the figure). The sequence ITS-1 contained a partial 16S rRNA gene (825 nt) nearly identical with a KSB3 16S rRNA gene previously reported as the bulking phylotype (clone YM-1, AB218870, Yamada et al., 2007). All of the 16S rRNA gene sequences obtained as ITS-1 from the previous bulking and non-bulking samples has one base mismatch with the clone YM-1, indicating the mismatch base in clone YM-1 may be a sequencing error introduced in PCR and cloning in the previous study (Yamada et al., 2007). The sequence ITS-1 contained two tRNA genes in the ITS region. The sequence ITS-2 had a partial 16S rRNA gene (824 nt) which is identical to a KSB3 16S rRNA gene previously obtained from the same reactor system (clone SmB78fl, AB266927). The sequence ITS-2 also contained two tRNA genes in the ITS region. These data show that both filament types are present in bulking and normally operating sludges.

Click here for additional data file.

Figure S3 Rank abundance and taxonomic affiliations of phylotypes found in the sludge samples

The aggregate top 100 OTU rank abundance for the four sludge samples (A1, A2, F1, G1) was generated using the 16S rRNA gene sequence close-reference OTU picking method in QIIME (http://qiime.org/tutorials/otu˙picking.html; QIIME v1.6.0; Caporaso et al., 2010a; Caporaso et al., 2010b; with the greengenes database (de-replicated dataset at 97%, March 2013, McDonald et al., 2012) as reference with the following parameters: otu_picking_method, uclust_ref; similarity cutoff value, 0.95) with all the shotgun paired-end metagenome data. Phylogenetic affiliation of phylotypes is color-coded by phylum. The UASB14 genome belongs to the second most abundant phylotype in the samples, whereas UASB270 belongs to the 43rd ranked phylotype.

Click here for additional data file.

Figure S4 Visualization of final population genome bins (A, UASB14 genome; B, UASB 270 genome)

Circular graphs from outside to inside: outermost circle with ticks for every 10 kbp (scale is shown as kbp) indicates scaffolds; CDS in forward strand (blue); CDS in reverse strand (blue), tRNAs (green) and rRNAs (red); the four outermost plots display G+C content (blue, from 0 [inner] to 100% [outer]), coverage with mate-pair reads from low [white], middle [blue], and high [red] coverages; innermost graph shows presence of broken-pairs (red) (longer than 15 kbp), end-links (green) and links with other scaffolds (blues) in mate-pair reads. Links for the broken-pairs (red), end-connections (green) and connections with other scaffolds (blues) in mate-pair reads are also shown in the circles.

Click here for additional data file.

Figure S5 Comparison of paired-end read coverage between KSB3 16S rRNA genes and genome scaffolds/contigs in relevant KSB3 genomes in the four metagenome data

The coverage of KSB3 16S rRNA gene sequence was estimated using the QIIME closed-reference OTU_picking_method (same as Fig. S3, except that relevant 16S rRNA gene sequences [YM-1, AB218870; SmB78fl, AB266927 ] were added into the greengenes database in the analysis). Coverage of genome scaffolds/contigs in the KSB3 genome bins was calculated as average coverage of all the associating scaffolds/contigs mapped with paired-end reads using BWA v0.7.4 with the BWA-MEM algorithm. (A) The coverage profile of the UASB14 16S rRNA gene and average coverage profile of all scaffolds/contigs in the UASB14 genome bin. The ratio of 16S to genome coverage was close to 3:1 indicating the presence of three copies of the rrn operon in a single genome. (B) The coverage profiles of UASB270 16S rRNA gene and all scaffolds/contigs in the UASB270 genome bin. The 16S to genome coverage ratio was ∼2:1 indicating the presence of two copies of the rrn operon.

Click here for additional data file.

Figure S6 Neighbor-joining tree based on 16S rRNA gene sequences showing the relative position of KSB3 (highlighted as red) amongst the major phylum-level lineages in the domain Bacteria

Ten 16S rRNA gene sequences representing the domain Archaea are used to root the tree. Taxonomic ranks are indicated by prefix; p__ (phylum), c__ (class). The phylogenetic robustness of each node is indicated by a symbol on the node: black circle (node was resolved in >90% of all the tree calculations including neighbor-joining, maximum parsimony, and maximum-likelihood inferences); open circle (resolved in >80% of all the calculations); open rectangle (resolved in >70% of all the calculations). The scale bar represents 10% estimated sequence divergence.

Click here for additional data file.

Figure S7 Maximum-likelihood phylogenetic inference of KSB3 population genomes among known bacterial phyla

The phylogenetic trees were constructed with RAxML (JTT and Gamma models with rapid 100 times bootstrapping, (A)) and FastTree (with default parameters, JTT model, CAT approximation, (B)) based on up to 38 marker genes using taxon-outgroup configuration ‘Config 1’, (Table S3). Sequences were collapsed at the phylum level except for KSB3 and Proteobacteria classes. KSB3 genomes obtained in this study are highlighted in red. Bootstrap resampling analysis was performed in each inference 100 times, and the values are displayed on interior nodes. The scale bars represents 5% inferred amino acid sequence divergence.

Click here for additional data file.

Figure S8 Histogram of pairwise BLASTP matches of UASB14 gene products to those of UASB270, Acidobacterium capsulatum ATCC 51196, Clostridium acetobutyricum DSM 1731, and Bacteroides fragilis 3_1_12

The number of genes sharing the same unit of amino acid identity are shown on the Y-axis. The average amino acid identity (AAI) for each genome pair are given in the figure legend.

Click here for additional data file.

Figure S9 Distribution (box plot) of genome size amongst bacterial and archaeal phyla estimated based on 2,279 finished genomes available in IMG (release 4.1, Markowitz et al., 2014).

The two representative KSB3 genomes (shown as red) obtained in this study currently have the largest average genomes amongst the phyla.

Click here for additional data file.

Figure S10 Distribution of cell envelope structure related genes in major bacterial phyla including KSB3

A maximum likelihood genome tree of the bacterial domain constructed using a concatenated alignment of up to 38 conserved proteins is shown at the left of the figure for phylogenetic ordering of traits shown in the heat map to the right. Black circles on interior nodes represent affiliations with >90% bootstrap support, and white circles represent branches with >80% support. Columns in the heat map represent individual gene families related to cell envelope biosynthesis (Albertsen et al., 2013), estimated using the annotation of finished bacterial genomes in IMG (release 4.1, Markowitz et al., 2014). Increasing representation of each gene family in a given phylum (percentage of genomes) is shown by increasing depth of color. Cell envelope classification is indicated by the abbreviations to the right of the phylum names: Monoderm (M), Diderm (D), Diderm-LPS (DL), Diderm-Atypical (DA).

Click here for additional data file.

Figure S11 Putative hydrogenase genes identified in the UASB14 and UASB270 genomes

Genes annotated as ‘hydrogenase’ are highlighted in red. In translated protein sequences from the hydrogenase genes, no sensory domains such as the PAS domain were found based on an InterProScan search (Quevillon et al., 2005). Some of the putative hydrogenase genes, however, are located in close proximity with some sensory genes such as protein kinase and PAS domain-containing proteins.

Click here for additional data file.

Figure S12 Distribution of the proportion of signal transduction genes amongst bacterial phyla

Distribution (box plot) of the proportion (% of the total ORFs) of signal transduction genes (COG category T) amongst bacterial phyla estimated based on the annotation of finished bacterial genomes in IMG (release 4.1, Markowitz et al., 2014). Two KSB3 genomes obtained in this study are highlighted in red, showing that they have amongst the highest proportion of signal transduction genes in the bacterial domain.

Click here for additional data file.

Figure S13 Normalized number (domains/Mb-genome) of domains related to environmental signaling systems identified in the two KSB3 genomes and 2,279 finished bacterial and archaeal genomes

Underlying data and labeling is the same as for Fig. 5. The normalized average signaling domain density and standard deviation are indicated by solid and dashed horizontal lines respectively, showing that the KSB3 genomes fall well outside the normal distribution.

Click here for additional data file.

Figure S14 Statistics of the gene systems contributing to “social IQ”

Statistics of the gene systems contributing to “social IQ” according to (Sirota-Madi et al., 2010) of the two KSB3 genomes (highlighted in red) and 2,279 reference bacterial and archaeal genomes from IMG (release 4.1, Markowitz et al., 2014). These comprise (A) two-component system genes, (B) transport related genes, (C) transcription factor genes, and (D) genes related to defense mechanisms. The combined scores are presented in (E) and (F), showing the high inferred social intelligence of the KSB3 bacteria relative to other prokaryotes.

Click here for additional data file.

Table S1 Shotgun sequencing statistics for UASB samples used for differential coverage banning, and relative representation of KSB3 genomes in thee data

Click here for additional data file.

Table S2 Summary of copy number anomalies found in the Modulibacteria (KSB3) genomes

Summary of copy number anomalies in the 111 conserved single copy marker gene set (Dupont et al., 2012) found in the Modulibacteria (KSB3) genomes and most parsimonious explanations based on phylogenetic inference of gene neighborhoods.

Click here for additional data file.

Table S3 Taxon-outgroup configurations for phylogenetic inference based on concatenated marker gene sets

Click here for additional data file.

Table S4 Support for Modulibacteria (KSB3) monophyly and affiliation with potential sister phyla based on phylogenetic inferences with varying taxon configurations, inference methods and marker gene sets

Click here for additional data file.

Table S5 16S rRNA-targeted fluorescence in situ hybridization (FISH) probes used in this study

Click here for additional data file.

Table S6 tRNA genes found in the Modulibacteria (KSB3) genomes

Click here for additional data file.

Table S7 Inventory of putative glycoside hydrolases (GHs) identified in the Modulibacteria (KSB3) genomes

Click here for additional data file.

Table S8 Inventory of Modulibacteria (KSB3) genes putatively involved in environmental signaling

Click here for additional data file.

Table S9 Bacteria ranked by proportion of encoded signaling proteins

Click here for additional data file.

Movie S1 Time-lapse video microscopy showing gliding motility of KSB3-like filaments in a UASB aggregate

The aggregate was exposed to a glucose gradient under anaerobic conditions at 37 °C. This video is shown at 45× real time.

Click here for additional data file.

We thank Jason Steen, Ben Woodcroft, Mohamed F. Haroon, and Michael Imelfort from the University of Queensland for assistance and advice on the bioinformatic analyses and Taeko Yokoi from AIST for assistance with ITS and FISH experiments. We also thank Satoshi Hanada from AIST and Bernhard Schink from the University of Konstanz for etymological advice.

Additional Information and Declarations

Competing Interests

Author Contributions

DNA Deposition

New Species Registration

Yuji Sekiguchi and Akiko Ohashi are employees of the National Institute of Advanced Industrial Science and Technology (AIST). Toshihiro Yamauchi is an employee of Kubota Kasui Corporation.

Yuji Sekiguchi conceived and designed the experiments, performed the experiments, analyzed the data, contributed reagents/materials/analysis tools, wrote the paper, prepared figures and/or tables, reviewed drafts of the paper.

Akiko Ohashi and Toshihiro Yamauchi performed the experiments, contributed reagents/materials/analysis tools, reviewed drafts of the paper.

Donovan H. Parks and Gene W. Tyson analyzed the data, contributed reagents/materials/analysis tools, reviewed drafts of the paper.

Philip Hugenholtz analyzed the data, contributed reagents/materials/analysis tools, wrote the paper, prepared figures and/or tables, reviewed drafts of the paper.

The following information was supplied regarding the deposition of DNA sequences:

The genome sequences of ‘Candidatus Moduliflexus flocculans’ and ‘Candidatus Vecturithrix granuli’ have been deposited in DDBJ/EMBL/GenBank under the accession numbers DF820455–DF820462 and DF820463–DF820483. The ITS sequences of the two genotypes have also been deposited in the databases under the accession numbers AB933567 and AB933568.

The following information was supplied regarding the registration of a newly described species:

Not applicable.

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
