# Peer review of "First genomic insights into members of a candidate bacterial phylum responsible for wastewater bulking"

_PeerJ, doi:10.7717/peerj.740_

## Round 0.1 · original submission · Minor Revisions

Your manuscript was reviewed by three peers, two of whom recommended "minor revisions," and one of whom recommended "major revisions." Therefore, I am recommending "minor revisions." I believe that this manuscript presents some interesting findings but that it would benefit from some careful revisions. The specific comments and constructive criticisms provided by the three reviewers fall under three main categories for the most part:

1) open access: submitting any scripts and raw data to public repositories so that others may use them;

2) clarifications for technical details: software versions and analysis parameters used, description of DNA extraction procedure, description of procedure for extracting 16S rRNA reads, etc.; and

3) considerable tightening up of the text: differentiation between where previous studies left off and this one began, grammar, typos, and careful word choice. Please pay careful attention to the specific criticisms of all the reviewers, including reviewer #3, who has made very specific suggestions to improve the text.

Reviewer 1 ·

Basic reporting

This manuscript is clearly written and well organized.

The values of the manuscript include:
1. The use of differential coverage binning of metagenomic data (a published method; GroopM) to reconstruct genomes from metagenomic sequences. This approach provides additional examples to the use of the program GroopM for analyzing deeply-sequenced metagenomics data, and extracting biological information from the inferred functions.
2. Biological insights into bacterial metabolic potentials through the analysis of genomic data, especially in a complex system where most of the community members are uncultivable.

The manuscript presents two high-quality draft genomes of an uncultivable bacterium KSB3, which was previously characterized in the bulking phenomenon of UASB technology based on 16S rRNA gene amplicon analysis. From the analysis of the reconstructed genomes, the authors proposed two new bacterial classes to describe the two KSB3 genomes, Moduliflexia and Vecturitrichia. The authors not only used the extracted 16S rRNA sequences to show that these two genomes belong to separate and previously unclassified clades, but also verified it with two sets of conserved marker genes found in other Bacteria and other domains of life. In addition, the authors used the genomic information to inferred bacterial metabolic properties and generated the motility hypothesis, which was tested through a small experiment.

Minor edits:

1. There is a typo at Supplementary Table 1: 250 bp instead of 250 b.

Experimental design

In general, the experimental design was well described. The value of this manuscript lies in the bioinformatics workflow. Although the methods are comprehensive and well presented in text, the readers will benefit from a graphical illustration of the workflow.

It is not clear whether the extraction of 16S rRNA reads from the metagenome datasets (line 113) were done using the raw single-end reads, or merged reads, or both, or the assembled reads.

Two sets of parameters were used (line 144) for SSPACE during the refinement and scaffolding of population genome bins with mate-pair reads. Explain why two sets of parameters were used, provide comparison of the resultant scaffold, and indicate which one was selected for the final scaffold.

It seems like the perl script “CompareToBootstrap.pl” was an in-house perl script. In keeping with true open access, these scripts, with a short description of their usage and code documentation should be made available.

Validity of the findings

The results, figures, and tables are well presented and discussed. An additional coverage plot illustrating the depth of coverage across both genomes would be helpful, especially for validating the accuracy of the assembly and scaffolding.

·

Basic reporting

No comment

Experimental design

No comment

Validity of the findings

No comment

Additional comments

Summary
In this manuscript the authors describe binning and analysis, using sophisticated bioinformatics, of two genomes from organisms belonging to candidate phylum KSB3. These filamentous organisms have been identified as a causing agent in sludge bulking of an UASB reactor. The genomes are used to construct a metabolic model of the organism, and gain insight in its ecophysiology. Additionally, based on the high amount of signal processing genes in both genomes, the authors hypothesize these organisms must be mobile and they proceed to show gliding motility using microscopy.

Although I think the study is well done and the manuscript is well written, there are a few things I’d like the authors to address:

In the methods section the DNA extraction procedure should briefly be mentioned, since it provides the underlying material for the data generated. Therefore, referring to a previous study (which in turn refers to a previous study) is in my opinion not right.

Accession codes are provided for the assembled/scaffolded genomes, but I could not find the raw data. The same goes for the study in which most of the sequencing was originally reported. I’d like to see the underlying raw data submitted to NCBI/EBI/DDBJ

Since UASB14 is the second most abundant organism of a non-bulking UASB it might benefit the paper to briefly discuss its role in the healthy system, and maybe speculate why it is more successful than UASB270.

Although in the methods section a range of stimulants for motility are mentioned, only glucose and maltose are mentioned in the results/discussion. In my opinion the discussion would be more complete when thoughts on the absence of response to the other stimulants are given.



in addition to the point above, I have a few minor comments on specific points in the text.

Line 46:
“cellular processes, including bulking,” seems to imply bulking is a cellular process. Maybe “cellular processes, including those causing bulking,” fits better?

Line 83:
Although I understand what is ment with population genomes, I think it is not (yet) an established term. Perhaps explaining the term in a few words will help establishing it quicker.

Line 175-176:
It is unclear what is ment with “a minimum similarity of 98% of the read length”, since the CLC mapper allows specification of ‘minimum similarity’ and ‘fraction of read length’ as separate parameters


Line 298:
The N50 for the assembly is given, but this metric is, in my opinion, quite meaningless for a metagenome since the sampling is (almost) always partial. I would advice to remove it.

Line 385:
the authors mention CRISPRs are present in all Archaea, but I think this is not true. Unless I’m mistaken they are absent from at least some thaumarchaea

Line 402:
The authors state the organisms generate ATP by converting acetyl-CoA to acetate. I’d add something along the lines of “in addition to glycolysis” as is shown in figure 4

Table 1:
There is some inconsistency in the notation where number of ORFs in cellular processes are mentioned. At “glycoside hydrolases” a percentage is given, then it is explained at “protease/peptidase”, and absent at “signalling”

·

Basic reporting

The submitted manuscript entitled “First genomic insights into members of a candidate bacterial phylum responsible for wastewater bulking” was reviewed. Authors of the publication by Yamada et al., 2011obtained four samples from UASB reactor sludge. Two time-points representing each flocculent sludge and granular sludge. Authors of the publication Soo et al. 2014 isolated DNA from these samples, sequenced DNA and perfomed metagenome analysis mainly focussing on cyanobacteria.

Authors of the currently manuscript also sequenced DNA from this isolated DNA. Authors of the publication Soo et al. 2014 stored the UASB Sludge samples in -80C in order to preserve the samples to perform FISH analysis later on. Population genomes of filamentous bacteria that belong to the candidate bacterial phylum KSB3 were obtained by genome binning. Both genomes were analysed for putative metabolic and sensory capacities. Phylogenetic relationships with other known micro-organisms was examined. Fish probes were designed based on 16S rRNA gene regions specific for the Modulibacteria KSB-3 phylum. To distinguish between the genome bins UASB14 and UASB270 specific helper probes were designed. Fish probe imaging revealed that both filamentous microorganisms are present in the relative abundance that was predicted from the sequencing data

Basic reporting:
English writing
The article should be overall improved on English writing. The choice of several English words (often verbs) should be re-considered. A couple of examples are:
Sentence 31 and s453 “gliding” motility. Please re-consider. I am not sure if gliding the best term to be chosen here. In think it is more about chemotaxis.
S33 “attuned”
S33-34 The terms “bulking” and “external cues” should be better explained in abstract but also in the rest of the manuscript.
S40 “anaerobic biotechnology” is not a type of treatment. “anaerobic digestion” is.
S47 “from the” The authors might mean “due to” in this case
S60 “remediate” I expect “prevent” is a better term here
S63 “are beginning to provide clues regarding” can be improved
S66 “high-quality draft population genomes” sounds very contradicting “high-quality” against “draft”so please explain this in the text.
S69 “constitute”
S79 “fractionated” should be more like “divided”
S81”fixed” is not the right term
S89”improve”
S89 “conducted” is not the right word
S93”CCD” should be explained
S117 the quality filtered parameters should be better explained. Same in sentences 126, 144, 145, 151, 152
In S211, 123, and 140 no websites should be mentioned in the text. The bioinformatic tool name, version and eventually date of last up-date should be mentioned with a correct reference to the scientific publication.
In the part between S127 and 152 the right terms should be used for scaffold or contig. As far as I understand they are not the same.
S176, 178 “beautified” is not a scientific term
S185-187 The sentence between brackets it too long
S186 “the another” is not correct English
S263 “broadly”
S298-300 This sentence should be written differently
S311 “verify the status” is unclear what is meant
S313 “assignment” reconsider
S323 “noting”.
S325 “general features common to the KSB3 genomes” “general” and “common” is the same.
S326 reconsider English in this sentence
S336-S338 is not clear
S401, S402, s442 “social intelligence” “social IQ”. Reconsider. “A high number of sensory systems” is scientifically more relevant and correct than “a high social IQ, or intelligence”
S457 “certain changes” is too vague and should be defined if used.
S552 “engaged in gliding”. The authors might mean “involved in chemotaxis”?
S604 typo, should be “Kleerebezem”

The authors should be consistent in the terms they use for UASB14 and UASB270 population genomes.

The authors should be consistent in the terms they use for the filamentous bacteria that belong to bacterial phylum KSB3. “KSB3”(s37), “KSB3 filaments”(s245), “KSB3 phylotypes”(S247), “bulking phylotype”(s250), clone YM-1. AB218870(s250). Choose one term. Explain the first time that you use it and use that term afterwards. Please reconsider the terms “phylotype” and “morphotypes”(s319).

Structure of the manuscript
The structure of the manuscript should be improved to some extent.

In the abstract FISH experiments should be mentioned.

In the introduction the authors should mention that previously Yamada et al, 2011 observed that when less glucose was fed to the reactor no bulking was observed for ...... days. Specify how many days. So this cannot be a conclusion from authors of the currently reviewed manuscript.
What is explained in the methods section s72-82 should be described in the introduction as this was performed by others.
The principle of the genome binning approach as previously described by Albertsen et al, 2013 should be clearly described in the introduction as this is key in this manuscript.
In the introduction the authors should shortly mention the methods they use to gain further insight in filamentous bacteria belonging to the phylum KSB3.

The result section starts with sentences that do not belong to the results section. From s243-247 until (Soo et al., 2014) should go to the introduction. It should be clear what was done by others and what is new.
S253-254 should go to the methods
S255-260 should go to the introduction. This part contains the more specified aim that was missing in the introduction.
S275-279 might fit better in the methods section
S290-293, why you do this 16S rRNA sequencing should be clearly mentioned in the methods section as well.
S364-367 starting from “fermentation....” should move to the introduction
S417-424 should move to the discussion

The first lines in the discussion section S426-433 would better fit in the introduction.

The conclusion should be improved. S475-479 is not a good conclusion (see further on “validity of the findings”)

Experimental design

S97-109 To the reviewer it was unclear what was the exact contribution of the sequences generated by Soo et al, 2014 on one hand and the sequences generated by the authors of the currently reviewed manuscript on the other hand. Were the sequences from Soo et al, 2014 only used for scaffolding? Were only data from sample A1 used? Was it really necessary to do extra sequencing in order to obtain the population genomes?
In general it should be better described why four samples were used. What is meant with “sample”. Is it DNA or sludge? Did the authors extract DNA themselves? What is the difference between the two sample dates? Was there a difference in reactor performance between those time points?

Validity of the findings

The data presented in the manuscript are statistically sound and seem robust and well controlled. The detailed Supplementary data contribute to this.

The conclusion should be improved. S475-479 is not a good conclusion and should be more specified. Indeed the population genomes provide new insight, but the authors should describe what is that new insight. Yamada et al 2011 did some observations. Authors of the currently reviewed manuscript can point out the pathways and other substrates that probably can be used by these bacteria.
The observation that a high number of genes involved in sensory systems were found is very interesting, especially in combination with chemotaxis analysis. The authors should consider removing terms like “overall social intelligence” or “social IQ scores” as it does not add anything scientifically.
The authors stated that previous attempts to isolate these filaments failed. Now they revealed the population genomes they should be able to propose novel isolation strategies, if not, it should be stated why.

Additional comments

The strategy used by the authors to reveal novel population genomes of filamentous bacteria that cause UASB reactor bulking is novel and provides novel insight. English writing and the layout must be improved in order to make it understandable for the reader. It should be clear what is done by others and what is performed by the authors of this manuscript. A clear description of the aim at the end of the introduction and of the conclusion is currently not present, but would improve the readability and would help to get a clear message across.

---

## Round 0.2 · accepted · Accept

Thank you for taking the time to carefully edit through the manuscript based on the reviewers' comments. I think the manuscript has much improved.